# Learning Compositional Koopman Operators for Model-Based Control

**Yunzhu Li**[*]    **Hao He**[*]    **Jiajun Wu**    **Dina Katabi**    **Antonio Torralba**
MIT CSAIL    MIT CSAIL    MIT CSAIL    MIT CSAIL    MIT CSAIL

## Abstract

Finding an embedding space for a linear approximation of a nonlinear dynamical system enables efficient system identification and control synthesis. The Koopman operator theory lays the foundation for identifying the nonlinear-to-linear coordinate transformations with data-driven methods. Recently, researchers have proposed to use deep neural networks as a more expressive class of basis functions for calculating the Koopman operators. These approaches, however, assume a fixed dimensional state space; they are therefore not applicable to scenarios with a variable number of objects. In this paper, we propose to learn compositional Koopman operators, using graph neural networks to encode the state into object-centric embeddings and using a block-wise linear transition matrix to regularize the shared structure across objects. The learned dynamics can quickly adapt to new environments of unknown physical parameters and produce control signals to achieve a specified goal. Our experiments on manipulating ropes and controlling soft robots show that the proposed method has better efficiency and generalization ability than existing baselines.

## 1 Introduction

Simulating and controlling complex dynamical systems, such as ropes or soft robots, relies on two key features of the dynamics model: first, it needs to be *efficient* for system identification and motor control; second, it needs to be *generalizable* to a complex, constantly evolving environments.

In practice, computational models for complex, nonlinear dynamical systems are often not efficient enough for real-time control (Mayne, 2000). The Koopman operator theory suggests that identifying nonlinear-to-linear coordinate transformations allows efficient linear approximation of nonlinear systems (Williams et al., 2015; Mauroy & Goncalves, 2016). Fast as they are, however, existing papers on Koopman operators focus on a single dynamical system, making it hard to generalize to cases where there are a variable number of components.

In contrast, recent advances in approximating dynamics models with deep nets have demonstrated its power in characterizing complex, generic environments. In particular, a few recent papers have explored the use of graph nets in dynamics modeling, taking into account the state of each object as well as their interactions. This allows their models to generalize to scenarios with a variable number of objects (Battaglia et al., 2016; Chang et al., 2017). Despite their strong generalization power, they are not as efficient in system identification and control, because deep nets are heavily over-parameterized, making optimization time-consuming and sample-inefficient.

In this paper, we propose compositional Koopman operators, integrating Koopman operators with graph networks for generalizable and efficient dynamics modeling. We build on the idea of encoding states into object-centric embeddings with graph neural networks, which ensures generalization power. But instead of using over-parameterized neural nets to model state transition, we identify the Koopman matrix and control matrix from data as a linear approximation of the nonlinear dynamical system. The linear approximation allows efficient system identification and control synthesis.

The main challenge of extending Koopman theory to multi-object systems is scalability. The number of parameters in the Koopman matrix scales quadratically with the number of objects, which harms the learning efficiency and leads to overfitting. To tackle this issue, we exploit the structure of the

---

[*]indicates equal contributions. Our project page: http://koopman.csail.mit.edu

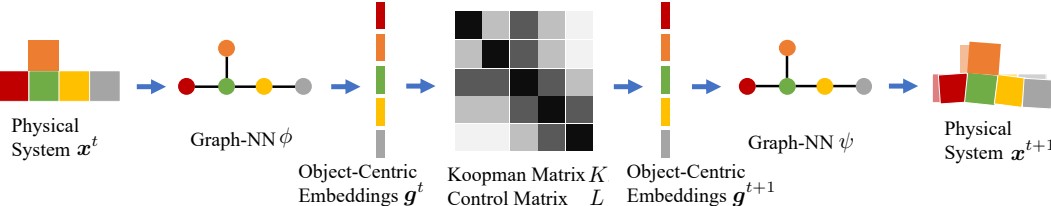

Figure 1: **Overview of our model.** A graph neural network $\phi$ takes in the current state of the physical system $\boldsymbol{x}^t$, and generates object-centric representations in the Koopman space $\boldsymbol{g}^t$. We then use the block-wise Koopman matrix $K$ and control matrix $L$ identified from equation 6 or equation 8 to predict the Koopman embeddings in the next time step $\boldsymbol{g}^{t+1}$. Note that in $K$ and $L$, object pairs of the same relation share the same sub-matrix. Another graph neural network $\psi$ maps $\boldsymbol{g}^{t+1}$ back to the original state space, i.e., $\boldsymbol{x}^{t+1}$. The mapping between $\boldsymbol{g}^t$ and $\boldsymbol{g}^{t+1}$ is linear and is shared across all time steps, where we can iteratively apply $K$ and $L$ to the Koopman embeddings and roll multiple steps into the future. The formulation enables efficient system identification and control synthesis.

underlying system and use the same block-wise Koopman sub-matrix for object pairs of the same relation. This significantly reduces the number of parameters that need to be identified by making it independent of the size of the system.

Our experiments include simulating and controlling ropes of variable lengths and soft robots of different shapes. The compositional Koopman operators are significantly more accurate than the state-of-the-art learned physics engines (Battaglia et al., 2016; Li et al., 2019b), and faster when adapting to new environments of unknown physical parameters. Our method also outperforms vanilla deep Koopman methods (Lusch et al., 2018; Morton et al., 2018) and Koopman models with manually-designed basis functions, which shows the advantages of using a structured Koopman matrix and graph neural networks. Please see our project page for demonstrating videos.

## 2 RELATED WORK

**Koopman operators.**   The Koopman operator formalism of dynamical systems is rooted in the seminal works of Koopman and Von Neumann in the early 1930s (Koopman, 1931; Koopman & Neumann, 1932). The core idea is to map the state of a nonlinear dynamical system to an embedding space, over which we can linearly propagate into the future. Researchers have proposed various algorithms to explore the Koopman spectral properties from data. A large portion of them are in the class of dynamic mode decomposition (DMD) (Rowley et al., 2009; Schmid, 2010; Tu et al., 2014; Williams et al., 2015; Arbabi & Mezic, 2017). The linear representation will enable efficient prediction, estimation, and control using tools from linear dynamical systems (Williams et al., 2016; Proctor et al., 2018; Mauroy & Goncalves, 2019; Korda & Mezić, 2018). People have been using hand-designed Koopman observables for various modeling and control tasks (Brunton et al., 2016; Kaiser et al., 2017; Abraham et al., 2017; Bruder et al., 2019b; Arbabi et al., 2018). Some recent works have applied the method to the real world and successfully control soft robots with great precision (Bruder et al., 2019a; Mamakoukas et al., 2019).

However, hand-crafted basis functions sometimes fail to generalize to more complex environments. Learning these functions from data using neural nets turns out to generate a more expressive invariant subspace (Lusch et al., 2018; Takeishi et al., 2017) and has achieved successes in fluid control (Morton et al., 2018). Morton et al. (2019) has also extended the framework to account for uncertainty in the system by inferring a distribution over observations. Our model differs by explicitly modeling the compositionality of the underlying system with graph networks. It generalizes better to environments of a variable number of objects or soft robots of different shapes.

**Learning-based physical simulators.**   Battaglia et al. (2016) and Chang et al. (2017) first explored learning a simulator from data by approximating object interactions with neural networks. These models are no longer bounded to hard-coded physical rules, and can adapt to scenarios where the underlying physics is unknown. Please refer to Battaglia et al. (2018) for a full review. Recently, Mrowca et al. (2018) and Li et al. (2019a) extended these models to approximate particle dynamics of deformable shapes and fluids. Flexible as they are, these models become less efficient during model adaptation in complex scenarios, because the optimization of neural networks usually needs a lot of

samples and compute, which limits its use in an online setting. Nagabandi et al. (2019a;b) proposed to use meta-learning for online adaptation, and have shown to be effective in simulated robots and a real legged millirobot. However, it is not clear whether their methods can generalize to systems with variable numbers of instances. The use of graph nets and Koopman operators in our model allows better generalization ability and enables efficient system identification as we only need to identify the transition matrices, which is essentially a least-square problem and can be solved very efficiently.

People have also used the learned physics engines for planning and control. Many previous papers in this direction learn a latent dynamics model together with a policy in a model-based reinforcement learning setup (Racanière et al., 2017; Hamrick et al., 2017; Pascanu et al., 2017; Hafner et al., 2019); a few alternatives use the learned model in model-predictive control (MPC) (Sanchez-Gonzalez et al., 2018; Li et al., 2019b; Janner et al., 2019). In this paper, we leverage the fact that the embeddings in the Koopman space are propagating linearly through time, which allows us to formulate the control problem as quadratic programming and optimize the control signals much more efficiently.

## 3 Approach

We first present the basics of Koopman operators: for a nonlinear dynamical system, the Koopman observation functions can map the state space to an embedding space where the dynamics become linear. We then discuss the compositional nature of physical systems and show how graph networks can be used to capture the compositionality.

### 3.1 The Koopman Operators

Let $\boldsymbol{x}^t \in \mathcal{X} \subset \mathbb{R}^n$ be the state vector for the system at time step $t$. We consider a non-linear discrete-time dynamical system described by $\boldsymbol{x}^{t+1} = F(\boldsymbol{x}^t)$. The Koopman operator (Koopman, 1931), denoted as $\mathcal{K} : \mathcal{F} \to \mathcal{F}$, is a linear transformation defined by $\mathcal{K}g \triangleq g \circ F$, where $\mathcal{F}$ is the collection of all functions (also referred to as *observables*) that form an infinite-dimensional Hilbert space. For every function $g : \mathcal{X} \to \mathbb{R}$ belonging to $\mathcal{F}$, we have

$$(\mathcal{K}g)(\boldsymbol{x}^t) = g(F(\boldsymbol{x}^t)) = g(\boldsymbol{x}^{t+1}), \tag{1}$$

making the function space $\mathcal{F}$ invariant under the action of the Koopman operator.

Although the theory guarantees the existence of the Koopman operator, its use in practice is limited by its infinite dimensionality. Most often, we assume there is an invariant subspace $\mathcal{G}$ of the Koopman operator. It spans by a set of base observation functions $\{g_1, \cdots, g_m\}$ and satisfies that $\mathcal{K}g \in \mathcal{G}$ for any $g \in \mathcal{G}$. With a slightly abuse of the notation, we now use $g(\boldsymbol{x}^t) : \mathbb{R}^n \to \mathbb{R}^m$ to represent $[g_1(\boldsymbol{x}^t), \cdots, g_m(\boldsymbol{x}^t)]^T$. By constraining the Koopman operator on this invariant subspace, we get a finite-dimensional linear operator $K \in \mathbb{R}^{m \times m}$ that we refer as the *Koopman matrix*.

Traditionally, people hand-craft base observation functions from the knowledge of underlying physics. The system identification problem is then reduced to finding the Koopman matrix $K$, which can be solved by linear regression given historical data of the system. Recently, researchers have also explored data-driven methods that automatically find the Koopman invariant subspace via representing the base observation functions $g(\boldsymbol{x})$ via deep neural networks.

Although the original Koopman theory does not consider the system with external control inputs, researchers have found that linearly injecting control signals to the Koopman observation space can give us good numerical performance (Brunton et al., 2016; Bruder et al., 2019a). Mathematically, considering a dynamical system, $\boldsymbol{x}^{t+1} = F(\boldsymbol{x}^t, \boldsymbol{u}^t)$, with an external control input $\boldsymbol{u}^t$, we aim to find the Koopman observation functions and the linear dynamics model in the form of

$$g(\boldsymbol{x}^{t+1}) = Kg(\boldsymbol{x}^t) + L\boldsymbol{u}^t, \tag{2}$$

where the coefficient matrix $L$ is referred to as the *control matrix*.

### 3.2 Compositional Koopman Operators

The dynamics of a physical system are governed by physical rules, which are usually shared across different subcomponents in the system. Explicitly modeling such compositionality enables more efficient system identification and control synthesis and provides better generalization ability.

**Motivating example.** Consider a system with $N$ balls moving in a 2D plane, each pair connected by a linear spring. Assume all balls have mass 1 and all springs share the same stiffness coefficient $k$. We denote the $i$'s ball's position as $(x_i, y_i)$ and its velocity as $(\dot{x}_i, \dot{y}_i)$. For ball $i$, equation 3 describes its dynamics, where $\boldsymbol{x}_i \triangleq [x_i, y_i, \dot{x}_i, \dot{y}_i]^T$ denotes ball $i$'s state:

$$\dot{\boldsymbol{x}}_i = \begin{bmatrix} \dot{x}_i \\ \dot{y}_i \\ \ddot{x}_i \\ \ddot{y}_i \end{bmatrix} = \begin{bmatrix} \dot{x}_i \\ \dot{y}_i \\ \sum_{j=1}^{N} k(x_j - x_i) \\ \sum_{j=1}^{N} k(y_j - y_i) \end{bmatrix} = \underbrace{\begin{bmatrix} 0 & 0 & 1 & 0 \\ 0 & 0 & 0 & 1 \\ k-Nk & 0 & 0 & 0 \\ 0 & k-Nk & 0 & 0 \end{bmatrix}}_{\triangleq A} \begin{bmatrix} x_i \\ y_i \\ \dot{x}_i \\ \dot{y}_i \end{bmatrix} + \sum_{j \neq i} \underbrace{\begin{bmatrix} 0 & 0 & 0 & 0 \\ 0 & 0 & 0 & 0 \\ k & 0 & 0 & 0 \\ 0 & k & 0 & 0 \end{bmatrix}}_{\triangleq B} \begin{bmatrix} x_j \\ y_j \\ \dot{x}_j \\ \dot{y}_j \end{bmatrix}.$$

(3)

We can represent the state of the whole system using the union of every ball's state, where $\boldsymbol{x} = [\boldsymbol{x}_1, \cdots, \boldsymbol{x}_N]^T$. Then the transition matrix is essentially a block matrix, where the matrix parameters are shared among the diagonal or off-diagonal blocks as shown in equation 4:

$$\dot{\boldsymbol{x}} = \begin{bmatrix} \dot{\boldsymbol{x}}_1 \\ \dot{\boldsymbol{x}}_2 \\ \vdots \\ \dot{\boldsymbol{x}}_N \end{bmatrix} = \begin{bmatrix} A & B & \cdots & B \\ B & A & \cdots & B \\ \vdots & \vdots & \ddots & \vdots \\ B & B & \cdots & A \end{bmatrix} \begin{bmatrix} \boldsymbol{x}_1 \\ \boldsymbol{x}_2 \\ \vdots \\ \boldsymbol{x}_N \end{bmatrix}.$$

(4)

Based on the linear spring system, we make three observations for multi-object systems.

- **The system state is composed of the state of each individual object.** The dimension of the whole system scales linearly with the number of objects. We formulate the system state by concatenating the state of every object, corresponding to an object-centric state representation.

- **The transition matrix has a block-wise substructure.** After assuming an object-centric state representation, the transition matrix naturally has a block-wise structure as shown in equation 4.

- **The same physical interactions share the same transition block.** The blocks in the transition matrix encode actual interactions and generalize across systems. $A$ and $B$ govern the dynamics of the linear spring system, and are shared by systems with a different number of objects.

These observations inspire us to exploit the structure of multi-object systems, instead of learning separate models for systems that contains different numbers of balls.

**Compositional Koopman operators.** Motivated by the linear spring system, we want to inject a good inductive bias to incorporate compositionality when applying the Koopman theory. This allows better generalization ability and more efficient system identification and better controller design. Figure 1 shows an overview of our model.

Considering a system with $N$ objects, we denote $\boldsymbol{x}^t$ as the system state at time $t$ and $\boldsymbol{x}_i^t$ is the state of the $i$'th object. We further denote $\boldsymbol{g}^t \triangleq g(\boldsymbol{x}^t)$ as the embedding of the state in the Koopman invariant space. In the rest of the paper, we call $\boldsymbol{g}^t$ the *Koopman embedding*. Based on the observation we made in the case of linear spring system, we propose the following assumptions on the compositional structure of the Koopman embedding and the Koopman matrix.

- **The Koopman embedding of the system is composed of the Koopman embedding of every objects.** Similar to the decomposition in the state space, we assume the Koopman embedding can be divided into object-centric sub-embeddings, i.e. $\boldsymbol{g}^t \in \mathbb{R}^{Nm}$ denoting the concatenation of $\boldsymbol{g}_1^t, \cdots, \boldsymbol{g}_N^t$, where we use $\boldsymbol{g}_i^t = g_i(\boldsymbol{x}^t) \in \mathbb{R}^m$ as the Koopman embedding for the $i$'th object.

- **The Koopman matrix has a block-wise structure.** It is natural to think the Koopman matrix is composed of block matrices after assuming an object-centric Koopman embeddings. In equation 5, $K_{ij} \in \mathbb{R}^{m \times m}$ and $L_{ij} \in \mathbb{R}^{m \times l}$ are blocks of the Koopman matrix and the control matrix, where $l$ is the dimension of the action for each object and $\boldsymbol{u}^t \in \mathbb{R}^{Nl}$ is the concatenation of $\boldsymbol{u}_1^t, \cdots, \boldsymbol{u}_N^t$ denoting the total control signal at time $t$:

$$\begin{bmatrix} \boldsymbol{g}_1^{t+1} \\ \vdots \\ \boldsymbol{g}_N^{t+1} \end{bmatrix} = \begin{bmatrix} K_{11} & \cdots & K_{1N} \\ \vdots & \ddots & \vdots \\ K_{N1} & \cdots & K_{NN} \end{bmatrix} \begin{bmatrix} \boldsymbol{g}_1^t \\ \vdots \\ \boldsymbol{g}_N^t \end{bmatrix} + \begin{bmatrix} L_{11} & \cdots & L_{1N} \\ \vdots & \ddots & \vdots \\ L_{N1} & \cdots & L_{NN} \end{bmatrix} \begin{bmatrix} \boldsymbol{u}_1^t \\ \vdots \\ \boldsymbol{u}_N^t \end{bmatrix}.$$

(5)

As we have seen in the case of linear spring system, those matrix blocks are not independent, but some of them share the same set of values.

- **The same physical interactions shall share the same transition block.** The equivalence between the blocks should reflect the equivalence of the interactions, where we use the same transition sub-matrix for object pairs of the same relation. For example, if the system is composed of $N$ identical objects interacting with the same relation, then, by symmetry, all the diagonal blocks should be the same, while all the off-diagonal blocks should also be the same. The repetitive structure allows us to efficiently identify the values using least squares regression.

### 3.3 LEARNING THE KOOPMAN EMBEDDINGS USING GRAPH NEURAL NETWORKS

For a physical system that contains $N$ objects, we represent the system at time $t$ using a directed graph $G^t = (O^t, R)$, where vertices $O^t = \{o_i^t\}_{i=1}^N$ represent objects and edges $R = \{r_k\}_{k=1}^{N^2}$ represent pair-wise relations. Specifically, $o_i^t = (x_i^t, a_i^o)$ , where $x_i^t$ is the state of object $i$ and $a_i^o$ is a one-hot vector indicating the object type, e.g., fixed or movable. For relation, we have $r_k = (u_k, v_k, a_k^r), 1 \leq u_k, v_k \leq N$, where $u_k$ and $v_k$ are integers denoting the end points of this directed edge, and $a_k^r$ is a one-hot vector denoting the type of the relation $k$.

We use a graph neural network similar to Interaction Networks (IN) (Battaglia et al., 2016) to generate object-centric Koopman embeddings. IN defines an object function $f_O$ and a relation function $f_R$ to model objects and their relations in a compositional way. Similar to a message passing procedure, we calculate the edge effect $e_k^t = f_R(o_{u_k}^t, o_{v_k}^t, a_k^r)_{k=1...N^2}$, and node effect $g_i^t = f_O(o_i^t, \sum_{k \in \mathcal{N}_i} e_k^t)_{i=1...N}$, where $\mathcal{N}_i$ denotes the relations that point to the object $i$ and $\{g_i^t\}$ are the derived Koopman embeddings. We use this graph neural network, denoted as $\phi$, to represent our Koopman observation function.

**System identification.**  For a sequence of observations $\widetilde{x} = [x^1, \cdots, x^T]$ from time 1 to time $T$, we first map them to the Koopman space as $\widetilde{g} = [g^1, \cdots, g^T]$ using the graph encoder $\phi$, where $g^t = \phi(x^t)$. We use $g^{i:j}$ to denote the sub-sequence $[g^i, \cdots, g^j]$. To identify the Koopman matrix, we solve the linear regression $\min_K \|Kg^{1:T-1} - g^{2:T}\|_2$. As a result, $K = g^{2:T}(g^{1:T-1})^\dagger$ will asymptotically approach the Koopman operator $\mathcal{K}$ with an increasing $T$. For cases where there are control inputs $\widetilde{u} = [u^1, \cdots, u^{T-1}]$, the calculation of the Koopman matrix and the control matrix is essentially solving a least squares problem w.r.t. the objective

$$\min_{K,L} \|Kg^{1:T-1} + L\widetilde{u} - g^{2:T}\|_2. \tag{6}$$

As we mentioned in the Section 3.2, the dimension of the Koopman space is linear to the number of objects in the system, i.e., $\widetilde{g} \in \mathbb{R}^{Nm \times T}$ and $K \in \mathbb{R}^{Nm \times Nm}$. If we do not enforce any structure on the Koopman matrix $K$, we will have to identify $N^2m^2$ parameters. Instead, we can significantly reduce the number by leveraging the assumption on the structure of $K$. Assume we know some blocks ($\{K_{ij}\}$) of the matrix $K$ are shared and in total there are $h$ different kinds of blocks, which we denote as $\hat{K} \in \mathbb{R}^{h \times m \times m}$. Then, the number of parameter to be identified reduce to $hm^2$. Usually, $h$ does not depend on $N$, and is much smaller than $N^2$. Now, for each block $K_{ij}$, we have a one-hot vector $\sigma_{ij} \in \{0,1\}^h$ indicating its type, i.e., $K_{ij} = \sigma_{ij}\hat{K} \in \mathbb{R}^{m \times m}$. Finally, as shown in equation 7, we represent the Koopman matrix as the product of the index tensor $\sigma$ and the parameter tensor $\hat{K}$:

$$K = \sigma \otimes \hat{K} = \begin{bmatrix} \sigma_{11}\hat{K} & \cdots & \sigma_{1N}\hat{K} \\ \vdots & \ddots & \vdots \\ \sigma_{N1}\hat{K} & \cdots & \sigma_{NN}\hat{K} \end{bmatrix}, \text{ where } \sigma = \begin{bmatrix} \sigma_{11} & \cdots & \sigma_{1N} \\ \vdots & \ddots & \vdots \\ \sigma_{N1} & \cdots & \sigma_{NN} \end{bmatrix} \in \mathbb{R}^{N \times N \times h}. \tag{7}$$

Similar to the Koopman matrix, we assume the same block structure in the control matrix $L$ and denote its parameter as $\hat{L} \in \mathbb{R}^{h \times m \times l}$. The least squares problem of identifying $\hat{K}$ and $\hat{L}$ becomes

$$\min_{\hat{K}, \hat{L}} \|(\sigma \otimes \hat{K})g^{1:T-1} + (\sigma \otimes \hat{L})\widetilde{u} - g^{2:T}\|_2, \tag{8}$$

where $\sigma \otimes \hat{K} \in \mathbb{R}^{Nm \times Nm}, \sigma \otimes \hat{L} \in \mathbb{R}^{Nm \times Nl}, g^{1:T-1} \in \mathbb{R}^{Nm \times (T-1)}$ and $\widetilde{u} \in \mathbb{R}^{Nl \times (T-1)}$. Since the linear least squares problems described in equation 6 and equation 8 have analytical solutions, performing system identification using our method is very efficient.

**Training GNN models.** To make predictions on the states, we use a graph decoder $\psi$ to map the Koopman embeddings back to the original state space. In total, we have three losses to train the graph encoder and decoder. The first term is the auto-encoding loss

$$\mathcal{L}_{\text{ae}} = \frac{1}{T} \sum_i^T \|\psi(\phi(\boldsymbol{x}^i)) - \boldsymbol{x}^i\|. \tag{9}$$

The second term is the prediction loss. To calculate it, we rollout in the Koopman space and denote the embeddings as $\hat{\boldsymbol{g}}^1 = \boldsymbol{g}^1$, and $\hat{\boldsymbol{g}}^{t+1} = K\hat{\boldsymbol{g}}^t + L\boldsymbol{u}^t$, for $t = 1, \cdots, T-1$. The prediction loss is defined as the difference between the decoded states and the actual states, i.e.,

$$\mathcal{L}_{\text{pred}} = \frac{1}{T} \sum_{i=1}^T \|\psi(\hat{\boldsymbol{g}}^i) - \boldsymbol{x}^i\|. \tag{10}$$

Third, we employ a metric loss to encourage the Koopman embeddings preserving the distance in the original state space. The loss is defined as the absolute error between the distances measured in the Koopman space and that in the original space, i.e.,

$$\mathcal{L}_{\text{metric}} = \sum_{ij} \left| \|\boldsymbol{g}^i - \boldsymbol{g}^j\| - \|\boldsymbol{x}^i - \boldsymbol{x}^j\| \right|. \tag{11}$$

Having Koopman embeddings that perserves the distance in the state space is important as we are using the distance in the Koopman space to define the cost function for downstream control tasks.

The final training loss is simply the combination of all the terms above: $\mathcal{L} = \mathcal{L}_{\text{ae}} + \lambda_1 \mathcal{L}_{\text{pred}} + \lambda_2 \mathcal{L}_{\text{metric}}$. We then minimize the loss $\mathcal{L}$ by optimizing the parameters in the graph encoder $\phi$ and graph decoder $\psi$ using stochastic gradient descent. Once the model is trained, it can be used for system identification, future prediction, and control synthesis.

## 3.4 CONTROL

For a control task, the goal is to synthesize a sequence of control inputs $\boldsymbol{u}^{1:T}$ that minimize $C = \sum_{t=1}^T c_t(\boldsymbol{x}^t, \boldsymbol{u}^t)$, the total incurred cost, where $c_t(\boldsymbol{x}^t, \boldsymbol{u}^t)$ is the instantaneous cost. For example, considering the control task of reaching a desired state $\boldsymbol{x}^*$ at time $T$, we can design the following instantaneous cost, $c_t(\boldsymbol{x}^t, \boldsymbol{u}^t) = \mathbb{1}_{[t=T]}\|\boldsymbol{x}^t - \boldsymbol{x}^*\|_2^2 + \lambda\|\boldsymbol{u}^t\|_2^2$. The first term promotes the control sequence that matches the state to the goal, while the second term regularizes the control signals.

**Open-loop control via quadratic programming (QP).** Our model maps the original nonlinear dynamics to a linear dynamical system. We can then solve the control task by solving a linear control problem. With the assumption that the Koopman embeddings preserve the distance measure, we define the control cost as $c_t(\boldsymbol{g}^t, \boldsymbol{u}^t) = \mathbb{1}_{[t=T]}\|\boldsymbol{g}^t - \boldsymbol{g}^*\|_2^2 + \lambda\|\boldsymbol{u}^t\|_2^2$. As a result, we reduce the problem to minimizing a quadratic cost function $C = \sum_{t=1}^T c_t(\boldsymbol{g}^t, \boldsymbol{u}^t)$ over variables $\{\boldsymbol{g}^t, \boldsymbol{u}^t\}_{t=1}^T$ under linear constrains $\boldsymbol{g}^{t+1} = K\boldsymbol{g}^t + L\boldsymbol{u}^t$, where $\boldsymbol{g}^1 = \phi(\boldsymbol{x}^1)$ and $\boldsymbol{g}^* = \phi(\boldsymbol{x}^*)$.

**Model predictive control (MPC).** Solving the QP gives us control signals, which might not be good enough for long-term control as the prediction error accumulates. We can combine it with Model Predictive Control, assuming feedback from the environment every $\tau$ steps.

## 4 EXPERIMENTS

**Environments.** We evaluate our method by assessing how well it can simulate and control ropes and soft robots. Specifically, we consider three environments. (1) **Rope** (Figure 2a): the top mass of a rope is fixed to a specific height. We apply force to the top mass to move it in a horizontal line. The rest of the masses are free to move according to internal force and gravity. (2) **Soft** (Figure 2b): we aim to control a soft robot that is consist of soft blocks. Blocks in dark grey are rigid and those in light blue are soft blocks. Each one of the dark blue blocks is soft but have an actuator inside that can contract or expand the block. One of the blocks is pinned to the ground, as shown using the red dots. (3) **Swim** (Figure 2c): instead of pinning the soft robot to the ground, we let the robot swim in fluids. The colors shown in this environment have the same meaning as in **Soft**.

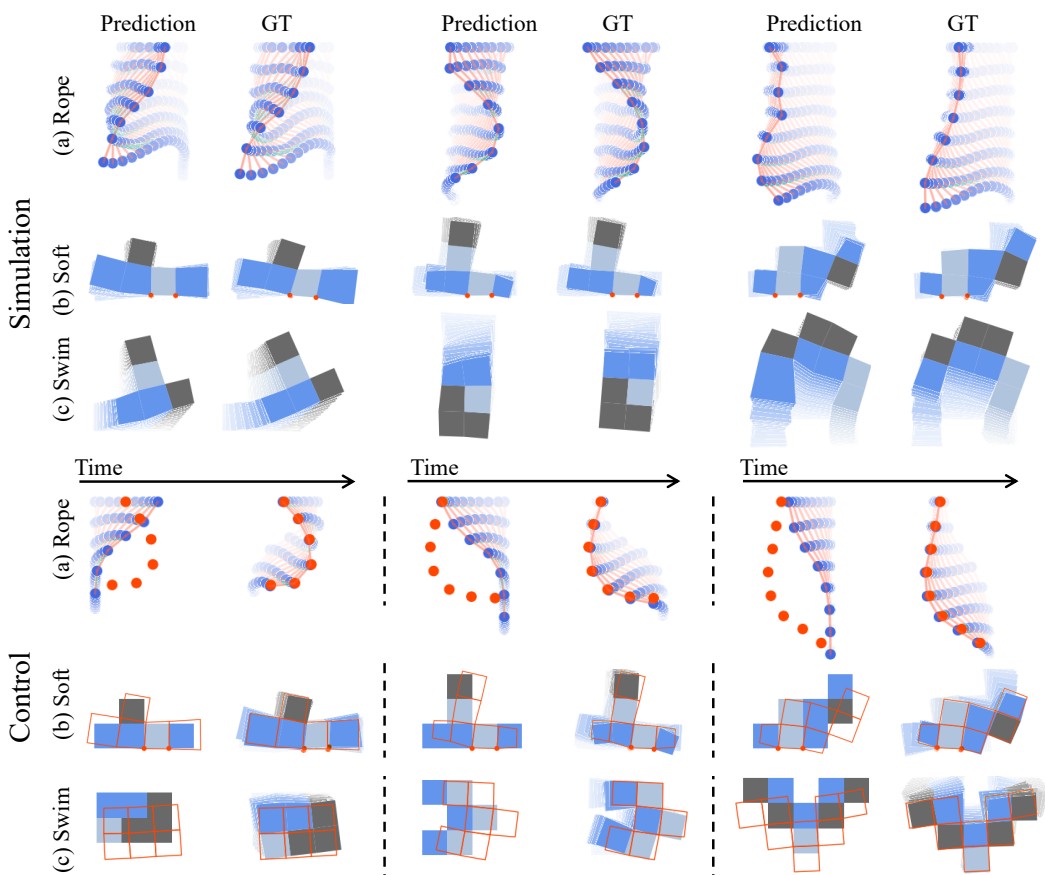

Figure 2: **Qualitative results.** Top: our model prediction matches the ground truth over a long period. Bottom: for control, we use red dots or frames to indicate the goal. We apply the control signals generated from our identified model to the original simulator, which allows the agent to achieve the goal accurately. Please refer to our supplementary video for more results.

**Observation space.** In the **Rope** environment, each mass on the rope is considered as an object. The observation of each mass is its position and velocity in the 2D plane, which has a dimension of 4. In total, a rope with $N$ masses has an observation space of dimension $4N$. In both the **Soft** and the **Swim** environments, each quadrilateral is considered as an object. For each quadrilateral, we have access to the positions and velocities of the four corners. Thus for a soft robot containing $N$ quadrilaterals, we have a $4 \times 4 \times N = 16N$ dimensional observation.

**Baselines.** We compare our model to the following baselines: Interaction Networks (Battaglia et al., 2016) (**IN**), Propagation Networks (Li et al., 2019b) (**PN**) and Koopman method with hand-crafted Koopman base functions (**KPM**). IN and PN are the state-of-the-art learning-based physical simulators, and we evaluate their adaptation ability by finetuning their parameters on a small sequence of observations from the testing environment. Similar to our method, KPM fits a linear dynamics in the Koopman space. Instead of learning Koopman observations from data, KPM uses polynomials of the original states as the basis functions. In our setting, we set the maximum order of the polynomials to be three to make the dimension of the hand-crafted Koopman embeddings match our model's.

**Data generation.** We generate 10,000 episodes for Rope and 50,000 episodes for Soft and Swim. Among them, 90% are used for training, and the rest for testing. Each episode has 100 time steps. In the dataset, the physical systems have a various number of objects from 5 to 9, i.e. the ropes have 5 to 9 masses while the soft robots in Soft and Swim environments have 5 to 9 quadrilaterals. To evaluate the model's extrapolating generalization ability, for each environment, we generate an extra dataset with the same size as the test set while containing systems consist of 10 to 14 objects.

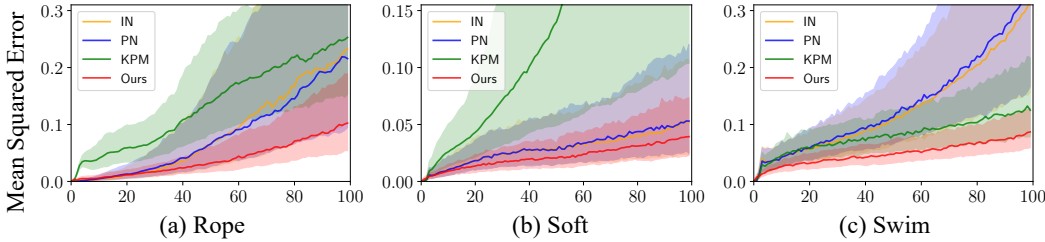

Figure 3: **Quantitative results on simulation.** The $x$ axis shows time steps. The solid lines indicate medians and the transparent regions are the interquartile ranges of simulation errors. Our method significantly outperforms the baselines in all testing environments.

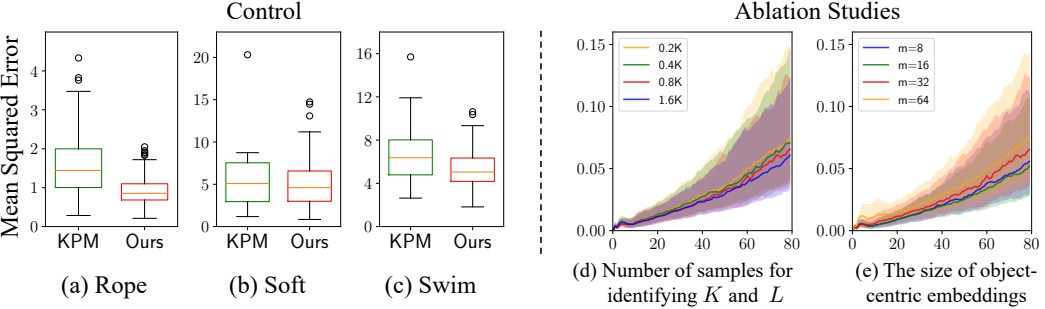

Figure 4: **Quantitative results on control and ablation studies on model hyperparameters.** Left: box-plots show the distributions of control errors. The yellow line in the box indicates the median. Our model consistently achieves smaller errors in all environments against KPM. Right: our model's simulation errors with different amount of data for system identification (d) and different dimensions of the Koopman space (e).

**Training and evaluation protocols.** All models are trained using Adam optimizer (Kingma & Ba, 2015) with a learning rate of $10^{-4}$ and a batch size of 8. $\lambda_1$ and $\lambda_2$ are 1.0 and 0.3, respectively, for our model. For both our model and the baselines, we apply 400K iterations of gradient steps in the **Rope** environment and 580K iterations in the **Soft** and **Swim** environment. Our model is trained on the sub-sequence of length 64 from the training set, and IN/PN aims at minimizing the L1 distance between their prediction and the ground truth. During test time, the models have to adapt to a new environment of unknown physical parameters, where they have access to a short sequence of observations and the opportunity to adjust their models' parameters. Our model uses 8 episodes to identify the transition matrix via least-square regression. IN/PN update the model's parameters by minimizing the distance between the model's prediction and the actual observation using a gradient step of length $10^{-4}$ for 5 iterations. For evaluation, we use two metrics: **simulation error** and **control error**. For a given episode, the simulation error at time step $t$ is defined as the mean squared error between the model prediction $\hat{x}^t$ and the ground truth $x^t$. For control, we pick the initial frame $x^0$ and the $t$'th frame $x^t$ from a episode. Then we ask the model to generate a control sequence of length $t$ to transfer the system from the initial state $x^0$ to the target state $x^t$. The control error is defined as the mean squared distance between the target state and the state of the system at time $t$.

## 4.1 SIMULATION

Figure 2 shows qualitative results on simulation. Our model accurately predicts system dynamics for more than 100 steps. For Rope, the small prediction error comes from the slight delay of the force propagation inside the rope; hence, the tail of the rope usually has a larger error. For Soft, our model captures the interaction between the body parts and generates accurate prediction over the global movements of the robot. The error mainly comes from the misalignment of some local components.

We evaluate the models by predicting 100 steps into the future on 500 trajectories and Figure 3 shows quantitative results. IN and PN do not work well in the Rope and Swim environments due to insufficient system identification ability. The KPM baseline performs poorly in the Rope and

Soft environments indicating the limited power of polynomial Koopman base functions. Our model significantly outperforms all the baselines.

## 4.2 CONTROL

We compare our model with KPM, the Koopman baseline using polynomial basis. In Rope, we ask the models to perform open-loop control where it only solves the QP once at the beginning. The length of the control sequence is 40. When it comes to Soft/Swim, each model is asked to generate control signals of 64 steps, and we allow the model to receive feedback after 32 steps. Thus every model has a second chance to correct its control sequence by solving the QP again at the time step 32.

As shown in Figure 2, our model leverages the inertia of the rope and matches the target state accurately. As for controlling a soft body swinging on the ground or swimming in the water, our model can move each part (the boxes) of the body to the exact target position. The small control error comes from the slight misalignment of the orientation and the size of the body parts. Figure 4 shows that quantitatively our model outperforms KPM, too.

## 4.3 ABLATION STUDY

**Structure of the Koopman matrix.** We explore three different structures of the Koopman matrix, *Block*, *Diag* and *None*, to understand its effect on the learned dynamics. *None* assumes no structure in the Koopman matrix. *Diag* assumes a diagonal block structure of $K$: all off-diagonal blocks ($K_{ij}$ where $i \neq j$) are zeros and all diagonal blocks share the same values. *Block* predefines a block-wise structure, decided by the relation between the objects as introduced in Section 3.3.

Table 1 includes our model's simulation error and control error with different Koopman matrix structures in Rope. All models are trained in the Rope environment with 5 to 9 masses. Besides the result on the test set, we also report models' extrapolation performance in parentheses, where the model is evaluated on systems with more masses than training, i.e., 10 to 14 masses.

Our model with *Block* structure consistently achieves a smaller error in all settings. *Diag* assumes an overly simplified structure, leading to larger errors and failing to make reasonable controls. *None* has comparable simulation errors but larger control errors. Without the structure in the Koopman matrix, it overfits the data and makes the resulting linear dynamics less amiable to the control.

Table 1: Ablation study results on the Koopman matrix structure (Rope environment). For simulation, we show the Mean Squared Error between the prediction and the ground truth at $T = 100$, whereas for control, we show the performance with a horizon of length 40. The numbers in parentheses show the performance on extrapolation.

|       | Simulation      | Control         |
|-------|-----------------|-----------------|
| Diag  | 0.133 (0.174)   | 2.337 (2.809)   |
| None  | 0.117 (0.083)   | 1.522 (1.288)   |
| Block | **0.105 (0.075)** | **0.854 (1.101)** |

**Hyperparameters.** In our main experiments, we set the dimension of the Koopman embedding to $m = 32$ per object. Online system identification requires 800 data samples for each training/test case. To understand our model's performance under different hyperparameters, we vary the dimension of the Koopman embedding from 8 to 64 and the number of data samples used for system identification from 200 to 1,600. Figure 4d shows that more data for system identification leads to better simulation results. Figure 4e shows that dimension 16 gives the best results on simulation. It may suggest that the intrinsic dimension of the Koopman invariant space of the Rope system is around 16 per object.

## 5 CONCLUSION

Compositionality is common in our daily life. Many ordinary objects contain repetitive subcomponents: ropes and soft robots, as shown in this paper, granular materials such as coffee beans and lego blocks, and deformable objects such as cloth and modeling clay. These objects are known to be very challenging for manipulation using traditional methods, while our formulation opens up a new direction by combining deep Koopman operators with graph neural networks. By leveraging the compositional structure in the Koopman operator via graph neural nets, our model can efficiently manipulate deformable objects such as ropes and soft robots, and generalize to systems with variable numbers of components. We hope this work could encourage more endeavors in modeling larger and more complex systems by integrating the power of the Koopman theory and the expressiveness of neural networks.

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

## A   ENVIRONMENT AND MODEL DETAILS

**Interaction types.**   In our experiments, interactions are considered different if the types are different or the objects involved have different physical properties.

In the **Rope** environment, the top mass has a fixed height and is considered differently from the other masses. Thus, we have 2 types of self-interactions for the top mass and the non-top masses. In addition, we have 8 types of interactions between different objects. The objects on a relation could be either top mass or non-top mass. It is a combination of 4. And the interaction may happen between two nearby masses or masses that are two-hop away. In total, the number of interactions between different objects is $4 \times 2 = 8$.

In the **Soft** environments, there are four types of quadrilaterals: rigid, soft, actuated, and fixed. We have four types of self-interactions correspondingly. For the interactions between objects, we add edges between two quadrilaterals only if they are connected by a point or edge. Connection from different directions are considered as different relations. There are 8 different directions, *up*, *down*, *left*, *right*, *up-left*, *down-left*, *up-right*, *down-right*. The relation types also encode the type of receiver object. Thus, in total, there are $(8 + 1) \times 4 = 36$ types of relations between different objects.

In the **Swim** environment, there are three types of quadrilaterals: rigid, soft, and actuated. Similar to the Soft environment, we use different edge types for different connecting directions; hence, the number of edge type is $(8 + 1) \times 3 = 27$.

## B   ADDITIONAL EXPERIMENTS

**Comparison with a classical physical simulator optimized using back-box optimization.**   We have performed comparisons with a classical physical simulator optimized using black-box optimization (Delingette, 1998) by assuming different levels of knowledge over the ground truth model.

If we assume that we know the ground truth model, where we only need to identify relevant physical parameters during the system identification stage, Bayesian Optimization (Snoek et al., 2012) (BO) can give us a reasonable estimate of the physical parameters. However, BO requires much more time to achieve a comparable performance with our method in the Rope environment: 0.43 vs. 180 seconds averaged over 100 trails (Ours vs. BO).

If we are unsure about the ground truth model and we approximate the system using a set of points linked by springs and dampers, BO does not work as well. In our additional experiments, we approximate the Rope environment using a chained spring-mass system, say $n$ masses and $n - 1$ springs. While taking much more time, BO still cannot give us a satisfying result: simulation error 0.046 vs. 0.084 and control error 0.854 vs. 2.547 (Ours vs. BO).

**Ablation study on the effectiveness of the metric loss.**   The internal linear structure allows us to solve the control problem using quadratic programming, where the objective function for control is defined in the embedding space (Section 3.4); hence, it is desirable to have Koopman embeddings that preserve the distance in the original state space. In Section 3.3, we introduce a metric loss to promote learning a Koopman embedding that keeps the distance measurement.

To demonstrate the effect of the metric loss, we compare the models trained with and without the metric loss. We establish the comparison using two measurements, the *distance preservation* and the *prediction accuracy*. To evaluate how well the Koopman embeddings preserve the distance, we compute the distribution of the log-ratio of the distance in the Koopman space and in the original state space, i.e., $\log \left( \frac{\|\boldsymbol{g}^i - \boldsymbol{g}^j\|_2}{\|\boldsymbol{x}^i - \boldsymbol{x}^j\|_2} \right)$. For the model prediction accuracy, we show the simulation errors.

We perform the experiments in the **Rope** environment and show the result in Figure 5. On the left, we show the ratio of distance in the learned Koopman space and the distance in the original state space. The model trained with metric loss has a log distance ratio that significantly more concentrates on 0. It means the metric loss effectively regularizes the model to preserve the distance. On the right, we show the simulation errors of the two models, which indicate that two models have comparable prediction performance. Metric loss effectively enhances the property of distance-preserving while not making a big sacrifice on the accuracy of the dynamics modeling.

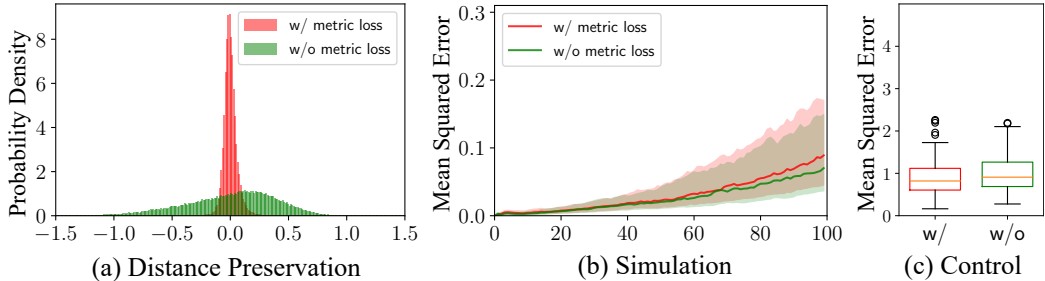

| (a) Distance Preservation | (b) Simulation | (c) Control |

Figure 5: **Ablation study on the metric loss in the Rope environment.** (a) shows the distributions of the logarithm distance ratio, i.e, $\log\left(\frac{\|\boldsymbol{g}^i - \boldsymbol{g}^j\|_2}{\|\boldsymbol{x}^i - \boldsymbol{x}^j\|_2}\right)$. The model trained with metric loss has a distance ratio much more concentrated to 1, which indicates it preserves the distance much better than the counterpart. (b) illustrates the simulation error of two models, where their performance is on par. (c) shows that the model trained using the metric loss performs better control.

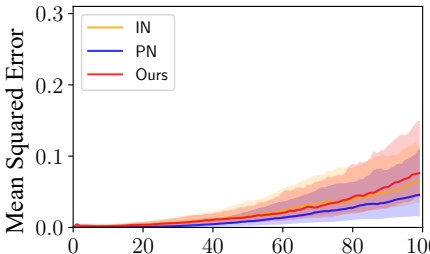

Figure 6: **Modeling rope with known physical parameters.** We show the comparison between our model and IN/PN in scenarios where we have access to the ground truth physical parameters. In this case, IN and PN slightly outperform our method due to the internal linear structure in our model. However, in the real world, we do not always know the physical parameters and their values, which makes our method preferable when adapting to new environments.

**Experiments in a known physical parameter setting.** Our setting is different from the settings in the original IN and PN papers that we do not assume we know the physical parameters and their values, such as stiffness, mass, and gravity. Instead, the parameters are embedded in the transition matrices during the system identification stage (Section 3.3). If the model has access to the underlying physical parameters, as expected, IN and PN slightly outperform our method as the internal linear structure limits our model's expressiveness, as shown in Figure 6. In the real world, however, the underlying physical parameters are not always known, which makes our model a better choice when adapting to unseen environments.

