# OpenReview forum: "Learning Compositional Koopman Operators for Model-Based Control"
_ICLR.cc/2020/Conference — Accept (Spotlight)_

### Official Review · AnonReviewer1 · 2019-10-19
**Official Blind Review #1**

**Rating:** 8

**Review:**

This paper introduces an approach to learning compositional koopman operators to efficiently model the dynamics of non-linear systems, consisting of an unspecified number of objects with repetitive dynamics.

The key contribution of this work is the use of a graph neural network that allows the koopman operator to be learned for systems of  multiple objects, and the incorporation of blockwise structure in the koopman gain and control matrices that improves parameter estimation process and is shown to reduce over-fitting.

Results show the proposed approach is effective, although only simple toy problems are examined and controlled. Nevertheless, this is a useful demonstration of learning for soft robot systems, and the idea of using koopman embeddings is likely to be of value to the ICLR community.

The paper is well written, and I like the idea of incorporating additional structure into the learning process through the expected blockwise structure.

As I understand it, the proposed approach is able to exploit this natural blockwise structure due to the assumption that the same physical dynamics are followed by each block (although objects can also be labelled as rigid/moving). How would this approach extend to objects with vary different properties (eg. 5 objects with connected with springs of different stiffness)?

Could you provide more details on the training process, is the model trained in an end-to-end fashion, or in parts?

For the control experiment, the choice was made to re-evaluate the control after 32 steps. Why was this the case? Is this due to the time taken to simulate/ find controls? MPC would typically re-plan faster than this.

On a related point, the introduction states that the proposed approach is 20 times faster than baselines, but no evidence of this is provided. How was this assessed, and is this at prediction or training time? How long does it take to replan using the proposed model and SQP?

**Experience Assessment:**

I have published one or two papers in this area.

**Review Assessment: Checking Correctness Of Derivations And Theory:**

I assessed the sensibility of the derivations and theory.

**Review Assessment: Checking Correctness Of Experiments:**

I assessed the sensibility of the experiments.

**Review Assessment: Thoroughness In Paper Reading:**

I read the paper at least twice and used my best judgement in assessing the paper.

---

> ### Author Response · Authors · 2019-11-13
> **Response to Reviewer #1**
>
> Thank you for your thoughtful and constructive comments!
>
> 1. Extending the approach to objects with different properties.
>
> Our current model can naturally handle objects with different properties by treating interactions having different physical properties as different interactions. For example, our model will treat springs with different stiffness as different interactions. It would be an interesting future direction to exploring formulations that can further reduce the parameter by grouping similar interactions. For example, the matrix blocks representing spring relations with different stiffness can share a similar structure.
>
> 2. Details on the training procedure.
>
> Our model is trained end-to-end. In every optimization step, we first use least-square regression to fit the Koopman matrix K and control matrix L, which are then used to calculate the loss functions. We then update the parameters in the graph encoder and decoder by backpropagating the gradient from the loss.
>
> 3. Details on the MPC procedure.
>
> In our MPC process, we re-evaluate the control after 32 steps. It is mainly due to the trade-off between the time used for control synthesis and the accuracy of the control result. In our three environments, our learned model is accurate enough that doing open-loop control gives reasonably accurate results. The benefits of replanning more frequently than once every 32 steps are marginal.
>
> 4. Computational time used for online adaptation.
>
> The statement in the introduction that “our model is 20 times faster when adapting to new environments of unknown physical parameters” is referring to the time used to do the online adaptation. The evaluation is performed in the Rope environment. Our model takes 0.43 +/- 0.11 seconds to identify the transition matrices using least-square regression while the IN/PN model needs 8.9 +/- 1.2 seconds for running 500 iterations of gradient descent using a learning rate of 1e-5. The statistics reported here are computed over 100 trials.
>
> Please let us know for any additional questions. Thanks!

---

### Official Review · AnonReviewer2 · 2019-10-28
**Official Blind Review #2**

**Rating:** 6

**Review:**

This paper proposes to learn compositional Koopman operators using graph neural networks to encode the state into object-centric embeddings and using a block-wise linear transition matrix to regularize the shared structure across objects.  The combination of deep Koopman operator with graph neural nets is very novel and interesting. The experiments are also well designed and the results shown in the paper also indicate the effectiveness and efficiency of the algorithm in both simulation and control. In conclusion, I think this work can inspire more wok into modeling larger and more complex systems by integrating the power of the Koopman theory and the expressiveness of neural networks.

**Experience Assessment:**

I do not know much about this area.

**Review Assessment: Checking Correctness Of Derivations And Theory:**

I assessed the sensibility of the derivations and theory.

**Review Assessment: Checking Correctness Of Experiments:**

I carefully checked the experiments.

**Review Assessment: Thoroughness In Paper Reading:**

I read the paper at least twice and used my best judgement in assessing the paper.

---

> ### Author Response · Authors · 2019-11-13
> **Response to Reviewer #2**
>
> Thank you for your thoughtful and encouraging comments! We are glad that you find our work novel, interesting and inspiring. We also appreciate that you find our experiments well designed and the result convincing.

---

### Official Review · AnonReviewer3 · 2019-11-03
**Official Blind Review #3**

**Rating:** 6

**Review:**

The paper is well written and the proposed idea is novel and builds on a sound theoretical framework of Koopman operator theory. A physical system is a represented as a graph; graph neural network is used to encode the current state to an object-centric embedding where the dynamics are assumed to be linear (Koopman operator theory) and modeled as a transition matrix. The key contribution is to recognize that similar physical interactions can be modeled using same parameters which constraints the transition matrix to be block-wise with shared parameters. Furthermore, the model is extended to add a control matrix to model external control. Experiments are conducted on simulations as well as control for 3 different settings - a a hanging rope/string anchored at the top, soft robot with an anchor, and soft robot in fluids.

Strengths
* Proposed method builds on a sound theoretical basis; although the linear dynamics model appear somewhat limited compared for the complex dynamics, thorough experiments are conducted to demonstrate the effectiveness of the method. The efficiency of the proposed algorithm compared to prior work makes is much practically useful.
* Well written with examples and illustrative figures.
* Quantitative analysis together with ablation studies on the structure are insightful.

Weaknesses
* My primary concern are with the evaluation.
- Experiments are only conducted on synthetic datasets. While experiments on real datasets are understandably difficult especially for quantitative validation, it would help to map the experiments to real problems to gain a more intuitive understanding and thus cater to a broader community.
- This paper and several related works are all evaluated on different problems; it would be useful to evaluate on similar tasks; for instance, strings [Battaglia 2016]. It would make it easier to draw comparisons.
- It's clear that the community would benefit significantly by having a benchmark or a web-based evaluation methodology (similar to OpenAI gym used actively reinforcement learning community). Unfortunately, this paper does not seem to offer a solution to this issue but continue to evaluate in ways similar to previous papers.


**Experience Assessment:**

I have read many papers in this area.

**Review Assessment: Checking Correctness Of Derivations And Theory:**

I assessed the sensibility of the derivations and theory.

**Review Assessment: Checking Correctness Of Experiments:**

I assessed the sensibility of the experiments.

**Review Assessment: Thoroughness In Paper Reading:**

I read the paper at least twice and used my best judgement in assessing the paper.

---

> ### Author Response · Authors · 2019-11-13
> **Response to Reviewer #3**
>
> Thank you for your thoughtful and constructive comments.
>
> 1. Real-world experiments
>
> We agree that showing real-world experiments would be beneficial. As a first step, we are starting with synthetic environments, which allow us to systematically evaluate and ablate on our model to fully understand its capability. We will explore ways to extend our model to the real world in the future.
>
> 2. Comparisons
>
> In our experiment, we make our environments as close to those used in the related works as possible. For example, our Rope environment is very similar to “string” in [Battaglia 2016]. We cannot directly use the same environment, because of the different problem setups: we study modeling and controlling a system with unknown physical parameters, while they focus on predicting future physical states given physical parameters.
>
> 3. Platform
>
> We also agree that building a platform such as the OpenAI gym that can benchmark different methods in the same environments would be valuable for the whole community. It can help to ensure that we are making concrete progress. As establishing such a benchmark has been beyond the focus of this submission, we leave it as future work.

---

### Official Review · AnonReviewer4 · 2019-11-04
**Official Blind Review #4**

**Rating:** 6

**Review:**

The paper proposes a novel method for modelling dynamical systems over graphs. The main idea investigated by the authors is to combine Graph Neural Networks together with approximate Koopman embedding. The GNN encodes the input graph to what the authors call "object-centric embedding", whose concatenation over all objects is defacto the approximate Koopman embedding of the system.
One of the key contributions is the reduction in parameters, by assuming that the interactions between different objects in the Koopman space are limited to some fixed number of types, or in other words given the object-centric embedding the Koopman matrix is a block matrix, where each block can only be one of K matrices. In this way the number of parameters is fixed and does not scale with the number of objects, compared to the naive way where it will scale as N^2. In addition to the dynamical modelling the paper adds an extra linear-"control" input in the Koopman embedding space which to affect the dynamics of the system and allow for modelling systems where there is external control being applied. The models are than compared on three small scale tasks, showing better results in mean squared error prediction compared to the three baseline approaches. Additionally, when used for controls on the environments the methods outperforms the one baseline method it is compared to.


I'm quite borderline on whether the paper should be accepted or rejected, but currently I'm leaning towards a rejection. The main reason for this decision is that in my opinion the experiments presented are somewhat limited with respect to the baselines used and I have some reservations regarding the results presented for IN and PN discussed below.


Detailed comments on paper:

1. I personally like the main idea of the paper, which is to use previous results from approximating the Koopman operator and combining it with GNNs for more accurate physical modelling of object-object interactions. Additionally, the idea of reducing the parameters is quite important.

2. Linear control theory - although it is quite natural to add the control as a linear affect in the latent space, and this has been done numerous times before in the literature, I don't recall there to be any theory on Koopman embedding when there is a control signal. Additionally, if the policy that has been used in practice is stochastic, the resulting "induced" dynamical system then also becomes stochastic, and to my knowledge, at least in theory, learning Koopman embedding for such systems has more challenges are requires certain assumptions about the true system, such as co-diagonalization and few others. I think this should be discussed in more detailed by the authors (and please do correct me if I'm wrong on any of these statements) as currently for the readers who are not too familiar I think the text might come across as though that the Koopman theory extends naively to these scenarios as well, which I do not think is the case.

3. It is not very clear how does the "metric" loss affects the solution. I would encourage the authors to provide comparison (only in terms of dynamical modelling, without active control) of whether this metric helps, or have some negative effects on the prediction. I think that for instance if the GNN has some form of weight regularization than this indeed would have some non-trivial effect on the resulting representation. Also, it would useful to have plots of how accurately does the embedding preserve the distance to the true states in order to understand this better.



Comments on the experiments:

1. In the paper there is no discussion about what are the actual observation space of the environments, could these be clarified better.

2. The block diagonal structure approach in general has been presented as working with multiple types of interactions. However, in practice it seems that the authors have only used two types of interaction -> object-same-object and object-other-object interaction. This however, has never been discussed and is maybe false. Could you clarify these details?

3. The results showed in Figure 3 are somewhat in contrast than the results in the original PN paper, specifically the PN paper states that it can achieve MSE of 7.85 for 1000 time steps, and from figure 6 of that paper it shows about 0.05 MSE over 100 steps on a similar rope environment. These results compared to the one presented here in Figure 3 makes me wonder how well did the authors actually managed to reimperilment the IN and PN paper? Could there be any comments on this as this makes many claims of the proposed method being better questionable and hard to understand its significance in relation to previous work.

4. For the control tasks, it would have been useful to have more than just the single baseline used. There are plenty of algorithms for Reinforcement Learning that could have been used in order to put the method in perspective. E.g. one can apply MPC with ground-truth model (e.g. the simulator) to show the discrepancy with an ideal case. In the RL literature there are plenty of methods for solving smaller problems, parametric and non-parametric: Q-learning, PPO etc... I think this is very important from the reader perspective.


PS: Please refer to the discussion below with the authors as I have increased my score from 3 to 6.

**Experience Assessment:**

I have read many papers in this area.

**Review Assessment: Checking Correctness Of Derivations And Theory:**

I assessed the sensibility of the derivations and theory.

**Review Assessment: Checking Correctness Of Experiments:**

I carefully checked the experiments.

**Review Assessment: Thoroughness In Paper Reading:**

I read the paper thoroughly.

---

> ### Author Response · Authors · 2019-11-13
> **Response to Reviewer #4**
>
> Thank you very much for your constructive and thoughtful comments,  and we would like to address your concerns as follows:
>
> 1. Ablation study on the effect of the metric loss.
>
> We have revised the paper and shown in Appendix B that the simulation performance is comparable with or without a metric loss. However, including metric loss effectively preserves the distance of the original state in the embedding space, which is a desirable character for control synthesis.
>
> Our objective function for control is defined in the embedding space (please see Section 3.4). The derived control signals aim at minimizing the L2 distance to the target embedding. Only when the distance is better preserved can minimizing the distance in the embedding space effectively minimizes the distance between states.
>
> In the newly added Figure 5(c), we show the control results in the Rope environment, which demonstrates that the model trained with metric loss has a better performance.
>
> 2. The correctness of the reimplementation of the baselines.
>
> Our setting is different from the settings in the original IN and PN papers since we do not assume we know the physical parameters and their values, e.g., stiffness, mass, gravity, etc. Instead, the parameters are embedded in the transition matrices during the system identification stage (Section 3.3).
>
> To check the fidelity of our reimplementation of IN/PN, we test our implementation in settings that are similar to the original IN/PN papers where the physical parameters are known. As shown in Appendix B in our revised paper, the simulation errors of our reimplementation are consistent with the errors reported in the original PN paper of around 0.05 at 100 time-steps. IN and PN slightly outperform our method as the internal linear structure limits our model's expressiveness. However, in the real world, we do not always know the physical parameters and their values, which makes our method preferable when adapting to new environments.
>
> 3. Observation space.
>
> In the Rope environment, each mass on the rope is considered as an object. The observation of each mass is its position and velocity, which has a dimension of 4. In total, a rope with N masses has an observation space of dimension 4N.
>
> In both the Soft and the Swim environments, each quadrilateral is considered as an object. For each quadrilateral, we have access to the positions and velocities of the four corners. Thus for a soft robot containing N quadrilaterals, we have a 4 * 4 * N = 16N dimensional observation.
>
> 4. Types of interactions.
>
> In our experiments, interactions are considered different if the types are different or the objects involved have different physical properties.
>
> In the Rope environment, the top mass has a fixed height and is considered differently from the other masses. Thus, we have 2 types of object-same-object interactions for the top mass and the non-top masses. In addition, we have 8 object-other-object interactions. The objects on a relation could be either top mass or non-top mass. It is a combination of 4. And the interaction may happen between two nearby masses or two masses that are two-hop away. In total, the number of object-other-object interactions is 4 * 2 = 8.
>
> In the Soft and the Swim environments, there are three types of quadrilaterals: rigid, soft and actuated. We have three object-same-object interactions correspondingly. For the object-other-object interactions, we add edges between two quadrilaterals only if they are connected by a point or edge. We are using different relation types if the types of the connected objects are different. In total, there are 3 * 3 * 2 = 18 object-other-objects.
>
> 5. The connection to linear control theory.
>
> The original Koopman theory paper did not consider dynamical systems with control. However, the lack of theory does not hamper the practical usage of the Koopman operator in many control tasks. Many previous papers [1, 2] followed the paradigm that adds the control as a linear effect in the latent Koopman space. While the optimality of these controllers has yet to be proved, “the numerical performance is striking” [1].
>
> The Koopman theory is developed for the deterministic dynamical system, which is also the focus of this paper. For the stochasticity introduced by the policy, i.e., we are uncertain about the action given the current state, our model can still work as long as the underlying dynamics are deterministic.
>
> We have revised the paragraph in section 3.1 to include the discussion.
>
> Please let us know for any additional questions. Thanks!
>
> [1] Steven L. Brunton, Bingni W. Brunton, Joshua L. Proctor, J. Nathan Kutz, “Koopman Invariant Subspaces and Finite Linear Representations of Nonlinear Dynamical Systems for Control,” PloS one 11.2 (2016): e0150171.
> [2] Daniel Bruder, Brent Gillespie, C. David Remy, Ram Vasudevan, “Modeling and Control of Soft Robots Using the Koopman Operator and Model Predictive Control,” in RSS 2019.

---

> > ### Comment · AnonReviewer4 · 2019-11-14
> > **Response to rebuttal**
> >
> > Firstly, thanks a lot for addressing many of the details I requested in my original review. I've read the updated version and I think the paper looks much better with these included.
> >
> > For me the weakest point of the paper still remains the fact that there does not seem to be enough baseline comparisons to other methods. This includes both in the dynamical modelling (e.g. there is plenty of literature on Deep Kalman Filters [1], SVAE [2] and follow up on those) and comparison to more standard RL algorithms for control. Of course, as this being research work I might be being a bit too harsh here.
> >
> > Nevertheless, because of the above reason I will increase my score to 6 rather than 8.
> >
> > [1] Rahul G. Krishnan, Uri Shalit, David Sontag, "Deep Kalman Filter"
> > [2] Matthew J. Johnson, David Duvenaud, Alexander B. Wiltschko, Sandeep R. Datta, Ryan P. Adams - "Composing graphical models with neural networks for structured representations and fast inference"

---

> > > ### Author Response · Authors · 2019-11-15
> > > **Thank you for your suggestions.**
> > >
> > > Thanks again for your suggestions, which have made the paper much stronger. We are glad to see that you find the paper now looks better. We’ll look into these additional baselines and hope to include them in the revised paper.
> > >
> > > Thanks!

---

### Author Response · Authors · 2019-11-13
**General Response**

We thank the reviewers for their constructive comments. We have revised the paper to address the concerns on the presentation and included two additional experiments in the appendix as suggested by the reviewers.

1. Ablation study on the effect of the metric loss.

2. Sanity check on the correctness of the reimplementation of the baselines.

3. Observation space and interaction types.

Please let us know if you have any questions. Thanks again for all the suggestions, which have made this submission stronger.

Best,
Authors.

---

### Decision · Program_Chairs · 2019-12-19

**Decision:**

Accept (Spotlight)

**Comment:**

This paper proposes using object-centered graph neural network embeddings of a dynamical system as approximate Koopman embeddings, and then learning the linear transition matrix to model the dynamics of the system according to the Koopman operator theory. The authors propose adding an inductive bias (a block diagonal structure of the transition matrix with shared components) to limit the number of parameters necessary to learn, which improves the computational efficiency and generalisation of the proposed approach. The authors also propose adding an additional input component that allows for external control of the dynamics of the system. The reviewers initially had concerns about the experimental section, since the approach was only tested on toy domains. The reviewers also asked for more baselines. The authors were able to answer some of the questions raised during the discussion period, and by the end of it all reviewers agreed that this is a solid and novel piece of work that deserves to be accepted. For this reason I recommend acceptance.